# Bioactive Inks Development for Osteochondral Tissue Engineering: A Mini-Review

**DOI:** 10.3390/gels7040274

**Published:** 2021-12-18

**Authors:** Negar Bakhtiary, Chaozong Liu, Farnaz Ghorbani

**Affiliations:** 1Department of Biomaterials, Faculty of Interdisciplinary Science and Technology, Tarbiat Modares University, Tehran 14115-114, Iran; negar.bakhtiary@modares.ac.ir; 2Institute of Orthopaedic & Musculoskeletal Science, University College London, Royal National Orthopaedic Hospital, Stanmore HA7 4LP, UK; chaozong.liu@ucl.ac.uk; 3Institute of Biomaterials, Department of Material Science and Engineering, University of Erlangen-Nuremberg, Cauerstraße 6, 91058 Erlangen, Germany

**Keywords:** bioactive ink, osteochondral, tissue engineering, scaffold, bioprinting

## Abstract

Nowadays, a prevalent joint disease affecting both cartilage and subchondral bone is osteoarthritis. Osteochondral tissue, a complex tissue unit, exhibited limited self-renewal potential. Furthermore, its gradient properties, including mechanical property, bio-compositions, and cellular behaviors, present a challenge in repairing and regenerating damaged osteochondral tissues. Here, tissue engineering and translational medicine development using bioprinting technology provided a promising strategy for osteochondral tissue repair. In this regard, personalized stratified scaffolds, which play an influential role in osteochondral regeneration, can provide potential treatment options in early-stage osteoarthritis to delay or avoid the use of joint replacements. Accordingly, bioactive scaffolds with possible integration with surrounding tissue and controlling inflammatory responses have promising future tissue engineering perspectives. This minireview focuses on introducing biologically active inks for bioprinting the hierarchical scaffolds, containing growth factors and bioactive materials for 3D printing of regenerative osteochondral substitutes.

## 1. Introduction

Osteochondral tissue has a complex structure consisting of articular cartilage and subchondral bone. An osteochondral defect arises from the damage in the cartilage and underlying bone, originating from an acute traumatic injury to the knee or bone disorder, leading to osteoarthritis (OA) [1]. According to the United Nations, by 2050, 15% of the world’s population will experience symptomatic OA or be disabled [2]. Based on the WHO, OA is estimated to affect >40 million people across Europe [3]. About 31 million Americans suffered from OA disease till 2012, and the population will rise to 65 million by 2030, making it essential to find a way to improve the quality of human life [4]. Hereon, tissue engineering (TE), a solution for repairing damaged tissues, is a promising approach for osteochondral defects treatment [5]. 

Scaffolds, the central part of TE, facilitate the growth and differentiation of the cells by creating a suitable environment for cell anchorage, spreading, and functionalization in the presence of signaling molecules, leading to tissue regeneration [6]. Herein, bioactive substances can bind to the surrounding tissue and prevent the activation of the immune system, leading to adaptation to the biological environment and accelerating the repair and regeneration of damaged tissue [7]. In the recent decade, advances in bioactive inks for 3D bioprinting scaffolds developed personalized implantable devices for bone and cartilage TE [8,9,10]. There are different classifications of bioprinting methods for osteochondral tissue regeneration. However, technically the three main bioprinting methods, including (1) inkjet-, (2) laser-, and (3) extrusion-based bioprinting, gained attention, considering bioink formula and required microstructure [11]. Herein, bioprinted scaffolds with the specific defect shape are implanted at the defect site and stimulate regeneration in the body’s environment. In a recent study, a bioactive bilayer bioprinted scaffold based on silk fibroin and decellularized extracellular matrix (ECM) enriched with osteogenic growth factors implanted in the rabbits’ knee joint model for osteochondral regeneration [12]. Besides, in-situ bioprinters that include a robotic arm and a handheld approach demonstrated particular application in bone and cartilage TE [13]. In an innovative experience, in-vivo hyaline-like cartilage formation was observed using in-situ bioprinting with coaxial nozzle extrusion-based technique based on gelatin methacrylate (GelMA) and hyaluronic acid (HA) [14]. Additionally, Keriquel and coworkers explored in-situ bone formation after implantation of a laser-based bioprinted scaffold based on collagen and hydroxyapatite (HAp) in a calvaria defect in a mice model [11].

This minireview concentrated on bioactive inks for bioprinting the personalized scaffolds for osteochondral tissue regeneration. We focused on introducing osteochondral tissue structure and properties. Then, general requirements of printable bioactive inks, such as rheological properties, printability, and shape fidelity, were explained. Eventually, bioactive inks containing relevant growth factors and bioactive materials for bone and cartilage regeneration are introduced as promising tissue repair approaches.

## 2. Osteochondral Tissue

Articular cartilage and subchondral bone make an osteochondral unit in the joints [15]. Articular cartilage, including the superficial, middle, and deep layers [16], is responsible for biomechanical properties of a joint such as distribution of force, absorption of shock, and load-bearing that enables a pain-free and frictionless movement of the joint [17,18,19]. The superficial layer with a parallel structure of collagens and a high density of chondrocytes protects against shear and tensile stresses [16,20]. More than that, the middle layer with randomly oriented collagens and chondrocytes bridge between the superficial and deep layers, absorbing the superficial layer stresses [21,22]. The deep layer resists against final stresses because of the unidirectional structure of chondrocytes and collagens [23]. Besides, the calcified layer facilitates fixation through biomineralization property [24]. 

The inflexible cortical layer, called cement line, separates the calcified cartilage from the subchondral bone and plays a remarkable role in nutrition transfer and vascularization [15,23]. Withal, subarticular spongiosa, the next part of the subchondral unit, provides epiphysis enriching with vessels, nerves, endothelium, and hematopoietic bone marrow [19]. Subchondral bone plate supports articular cartilage and decreases the loads transmitted to cartilage into 1–3% by absorbing 30% of stresses [25,26,27]. 

Osteochondral tissue injuries, caused by trauma or osteochondritis deformations, affect the natural tissue structure, such as reducing trabecular bone density and flexibility or increasing the thickness of the subchondral bone layer, especially in elderly patients [4,19,28]. On this point, based on the depth of injury, osteochondral defects are classified into five grades (Figure 1) based on the Outerbridge classification system, and there are 5 classification for osteochondral tissue: (0) normal cartilage; (1) mild lesion with just softening and swelling in cartilage; (2) moderate lesion with partial-thickness and width less than 0.5 inches; (3) severe lesion with full-thickness and width more than 0.5 inches; and (4) very severe lesion with subchondral bone damage, which means involvement of the entire thickness of the osteochondral tissue, about 2 to 4 mm [16,29,30,31]. Such damages can be repaired using multicomponent hierarchical and stratified scaffolds [23]. Due to the possibility of mimicking the gradient structure and properties of the osteochondral unit, bioprinting is a suitable method for personalized osteochondral tissue regeneration and translational medicine. The bioprinting method makes it possible to fabricate multilayer and multimaterial precise structures and resembling defect size and shape by using 3D imaging such as MRI (magnetic resonance imaging) or CT scan (computed tomography) in processing with CAD (computer-aided design) software [32].

## 3. The Requirements of Bioink

The cell-laden or cell-free synthetic or natural materials used to fabricate scaffolds in bioprinting technology are called bioink [33]. Many parameters should be considered in formulating a bioink, such as cell interactions with chemical reagents and cross-linkers, the effect of mechanical stresses during printing and deposition, nutrition accessibility for cells after printing, shape fidelity, and printability, which all affect the final properties, cell behavior and tissue regeneration [34]. Bioink’s rheological properties, flowability, and deformability are considered critical factors, determining the printability of the bioink, shape fidelity, and stability of the final structure [35,36,37]. 

One of the most crucial rheological properties of bioinks, shear-thinning, affects viscosity and is categorized into Newtonian and non-Newtonian fluids based on shear stress and shear rate behavior. Higher viscosity leads to better shape fidelity; in contrast, shear-thinning bioinks decrease the viscosity by the high shear rate, making excellent printability and reducing the injection force with a more negligible effect on cell viability [36,38]. 

Moreover, bioinks should have a viscoelastic property to overcome the surface’s tension, maintain each layer’s integrity, and support adhesion [39]. Due to having cell suspensions and the ability of elastic form restoring, dynamic storage modulus (G′) is essential by illustrating the storage energy of the ink in solid-like behavior. In addition, loss modulus (G″) and loss tangent δ (G″/G′) expose the printability and possibility of cell mixing in the liquid-like behavior of bioink [40,41]. 

Yield stress, the resistance force from the material against flowing, makes each printed layer withstand the next layer without changing the shape and preventing the sedimentation of various cells and additive components within the bioink [42]. Hereabouts, the dynamic yield stress is the lowest force for keep flowing, and static yield stress is the lowest force for commencing the flow during deposition [43]. The shape fidelity of the bioink is supplied when the applied force is lower than the yield stress and elastic properties. Nonetheless, upper than the yield point, the deformation required during the injection will start [42]. All the rheological and mechanical properties mentioned depend on the formulated bioink and applied bioprinting technology. 

## 4. Bioactive Inks

TE scaffolds are implanted in the defective site by surgery, which leads to the immune responses by a series of inflammatory activation immediately after the cell-implant interactions and is recognized as a foreign object [44]. Besides the counteracting inflammatory factors, macrophages can regulate protein expression for cell growth and regeneration [45,46]. Therefore, adding bioactive substances to implantable scaffold activates the immune system through binding to the relevant biological environment and facilitating the secretion of repair factors by macrophages, which increases the implantation’s success [31]. Accordingly, bioactive ink, which refers to bioink containing biologically active additives, found versatile applications for fabricating personalized osteochondral scaffolds via bioprinting technology [9]. This classification of printable inks increases bondability with soft or hard tissues, control immune responses, and stimulate differentiation to regenerate the defects. Table 1 portrays a summary of reviewed literature on bioactive inks.

Growth Factor-containing inks

Growth factors, diffusible signaling proteins, are a group of bioactive components versatile in the fabrication of bioactive inks [88,89]. Growth factor-laden bioinks stimulate cell proliferation, differentiation, vascularization, and tissue repair by bonding the growth factor receptors on the surface of target cells. A variety of growth factors support osteogenic and chondrogenic differentiation as well as regulate osteoblasts’ activity for bone regeneration. For instance, bone morphogenetic protein (BMP) (2, 4, 6, and 7), or members of transforming growth factor-β (TGF-β) family, support chondrocyte and osteogenic differentiation, bone and fracture healing, bone and cartilage development, bone marrow formation, etc. [90,91,92,93,94]. Alternatively, fibroblast growth factor-2 (FGF-2) promotes bone development and expresses at endochondral bone formation stages as well as regulating chondrocytes differentiation and proliferation [95]. Insulin-like growth factor (IGF) (1, 2) is another influential growth factor in osteogenic and chondrogenic differentiation [4,93,96,97,98,99,100]. Additionally, platelet-derived growth factor (PDGF) and vascular endothelial growth factor (VEGF) have demonstrated synergistic effects besides the factors mentioned above to increase osteogenic and chondrogenic differentiation and tissue regeneration [101,102,103,104,105,106]. On the other hand, interleukin-10 (IL-10), an anti-inflammatory cytokine involved in bone regeneration, can be incorporated into the formulation of bioactive inks [107]. 

Wang and colleagues formulated bioactive ink by interpenetrating the alginate-GelMA network incorporated with TGF-β3 [47]. They found that the release of TGF-β3 significantly promotes cartilage ECM deposition. A group of researchers fabricated two alginate-based bioactive inks, separately loaded with VEGF and BMP-2 (Figure 2A) [48]. The results illustrated that growth-factor-loaded bioactive inks improve angiogenesis or osteogenesis of bone marrow mesenchymal stem cells (BMSCs) and support new bone regeneration. Cooper et al. [49] modified DermaMatrix, a human allograft consisting of acellular dermal ECM, with BMP-2 and noggin to prepare bioactive ink for inkjet bioprinting. Here, the synergistic effect of BMP-2 and noggin exhibited an improvement in in-vitro osteogenic differentiation and in-vivo bone regeneration. A group of scholars used a bioprinting strategy to design a bioactive scaffold centering on bone TE [50]. Alginate bioinks incorporated with BMP-2-loaded gelatin microparticles were used to prepare scaffolds. Implantation of scaffolds in a mice model for six weeks confirmed the effectiveness of gelatin microparticles for controllable release of BPM-2, which leads to BMP-2 long-lasting and significantly supporting osteogenicity as well as promoting bone regeneration. Kundu and colleagues investigated the chondrocytes and TGF-β loaded polycaprolactone (PCL)-alginate inks for cartilage formation [51]. In-vitro results exhibited higher cartilage ECM formation based on higher glycosaminoglycans (GAGs), deoxyribonucleic acid (DNA), and total collagen contents. Implantation of printed bioactive inks in the dorsal subcutaneous space of mice revealed an increase in cartilage tissue and type II collagen fibril formation. In this regard, they concluded that growth factors addition to printable ink formulation positively affects cartilage tissue regeneration. Another study by a group of researchers exhibited that the addition of TGF-β3 to difunctional scaffold based on aptamer and decellularized cartilage ECM dispersed in GelMa promotes chondrogenic differentiation of BMSCs [52]. The improvement of full-thickness defect regeneration in rabbits’ cartilage demonstrated that the growth factor increment and aptamer could direct the cell, a promising strategy for in-situ cartilage regeneration.

As the common conventional routes are not beneficial due to the short half-life of growth factors in blood circulation, enriched-bioactive scaffolds with growth factors can be a promising solution due to keeping functionality, control release, and targeted performance of growth factors [4].

DNA-containing inks

DNA-contained inks are the other member of biologically active inks. DNA hydrogels are biodegradable, designable, and permeable compositions, constructing ingenious TE patterns [54]. Therapeutic proteins expression can stimulate tissue formation following ECM production, and this approach is achievable by nonviral gene delivery. In this regard, gene-activated inks provide a promising approach in biofabrication strategies and the biological functionality of scaffolds [56].

In a recent study, BMSCs-laden alginate-methylcellulose hydrogels enriched with osteogenic and chondrogenic genes encoding plasmid DNA (pDNA) for fabricating bioprinted multi-layer scaffold supporting osteochondral regeneration was developed [53]. The in-vivo results of gene-activated bioinks demonstrated promoting vascularization, as well as producing a stable cartilage layer over bony tissue after 28 days. Moreover, Li and coworkers studied cell-laden supramolecular polypeptide-DNA hydrogels for rapid fabrication of in-situ multi-layer scaffolds using 3D printing technology for TE applications [54]. Here, in-situ hydrogels were prepared by mixing polypeptide-DNA conjugate as bioactive ink A and a complementary DNA linker as bioactive ink B. DNA-based inks exhibited supreme healing properties owing to dynamic cross-linking by DNA hybridization. The viability of AtT-20 (an anterior pituitary cell line) and HEK-293 (Human embryonic kidney 293) was about 98% means the bioactive ink provides proper support for cells. Furthermore, printed scaffolds exhibited shape stability from high mechanical strength and non-shrinkage properties. In a parallel study, Loozen et al. [55] used alginate-pDNA containing BMP-2 encoded gene supplemented with BMSCs to investigate osteogenic differentiation and bone formation activating after in-vivo implantation. Bioactive ink indicated suitable printability and shape fidelity as well as mechanical properties. The elevated production of BMP-2 and alkaline phosphatase (ALP) portrayed osteogenic differentiation and efficient transfection of cells. After six weeks of scaffold implantation in the mice model, the hematoxylin and eosin staining illustrated partial degradation of the structure, which led to DNA release and subsequently provided the surrounding tissue access to BMP-2 genes. Higher availability of BMP-2 genes makes for faster osteogenic differentiation, which is proved by increased deposited collagen content. 

Additionally, A group of scholars investigated encapsulated BMSCs in alginate-nano-HAp bioactive ink containing pDNA with BMP-2 and TGF-β3 encoding genes [56]. The in-vitro results indicated deposition and mineralization of matrix due to sustained expression of pDNA and encoding genes after 14 days. In addition, the high levels of mineralization and vascularization were proved by the implantation (subcutaneously into the back of nude mice (Balb/c)) of gene-activated scaffolds for 12 weeks. Incorporating bioinks with DNA for bioprinting systems opens novel concepts obtained from molecular programming in the TE. The future potential of 3D-printed scaffolds applications or the advancement of scaffold fabrications can be used to immobilize DNA-tagged segments in biocompatible 3D conditions [108]. Given its stimulatory genetic codes for cell growth and differentiation, it is convincing that DNA could be a promising candidate as a bioactive component for regeneration and TE programs.

ECM-based inks

The importance of ECM in cell growth and differentiation on tissue repair is evident. It was demonstrated that decellularizing ECM can play a remarkable role in cellular activities in cartilage TE [109]. Decellularization of ECM effectively removes components of cells and DNA, but not entirely, which the remaining components such as cell membrane particles can affect chondrogenic differentiation ability [110]. 

In an innovative experience, a BMSCs-laden bioactive ink consisting of silk fibroins and decellularized ECM was fabricated [57]. According to observations, the matrix could support proliferation and chondrogenic differentiation as a function of decellularized ECM addition besides the good mechanical properties and degradation rate. In another recent study, a 3D structure was produced using engraving decellularized cartilage, called CartiScaff, derived through a CO_2_ laser of human articular cartilage [58]. Implantation of the scaffold in articular cartilage defect of mice osteochondral plugs illustrated increased cellular interaction and improved cartilage regeneration. Zhu and coworkers lately developed a novel osteochondral scaffold based on decellularized ECM-polyethylene glycol diacrylate (PEGDA) integrated hydrogel to evaluate the chondrogenic promotion BMSCs potential [59]. The implanted PEGDA-ECM scaffolds in rat osteochondral defect portrayed subchondral bone regeneration with few effects on cartilage repair. A group of scholars used a new approach based on released ECM from preosteoblast seeded cells in collagen gel after three days demonstrating significant bioactivity in promoting osteogenic differentiation and bone regeneration in alginate-collagen-ECM bioactive ink formula [60]. Due to its structural proteins, the ECM component provided cellular activities such as migration, growth, and differentiation by biochemical signals.

Depending on in-vitro and in-vivo results, the decellularized ECMs can serve as a beneficial bioactive component to promote regeneration. However, many factors such as genetic, species, age, or health status of the donors, in addition to specific disease state and zonal variety, affect its composition and functionality [111].

Bioactive polymer-based inks

Natural polymers are categorized in the group of bioactive materials due to molecular properties and have illustrated an excellent stimulation on the growth of chondrocytes, which effectively repair articular cartilage [112,113]. 

Collagen, a critical component of osteochondral ECM, can be considered a bioactive additive for osteochondral scaffolds [114]. Lee and colleagues fully justified the bioactivity of collagen by chondrocytes cultured in alginate-collagen bioink that could expose related chondrogenic markers [61]. The comparison of alginate-collagen scaffolds with alginate and alginate-agarose scaffolds portrayed that collagen-containing scaffolds have more chondrocyte phenotype maintenance. Furthermore, a group of researchers observed collagen type I potential for BMSCs osteogenic regeneration and mineralization in the bioprinted agarose-based scaffold [62]. Higher collagen concentration led to increased cell spreading enhancement and BMSCs osteogenic differentiation. On the other hand, the low mechanical properties of pure collagen make it necessary to combine with other materials for improving rheological properties and printability, such as alginate or agarose, as mentioned. Gelatin, as a product of collagen, has demonstrated bioactive properties too. On this point, in an innovative experience, a porous bilayered scaffold with GelMA and GelMA-HAp bioactive inks was fabricated, which can support osteochondral repair in 6–12 weeks [63]. Bioactivity originated from Arg-Gly-Asp (RGD) peptide sequence in gelatin chemical structure, stimulating cartilage ECM composition and making it suitable for cell adhesion.

HA, a glycosaminoglycan (GAG), provides molecular signaling for proliferation and migration, effective in tissue regeneration, but poor mechanical properties and printability have increased the need for modifying [115,116]. Antich et al. [64] stated that adding HA as one of the main components of cartilage to alginate bioink significantly increased chondrogenic markers expression, promoted chondrogenesis as well as provided suitable mechanical properties. Again, a group of scholars indicated the potential of bioactive norbornene-modified HA ink for chondrogenic differentiation of BMSCs in 56 days (Figure 2B) [65]. 

**Figure 2 gels-07-00274-f002:**
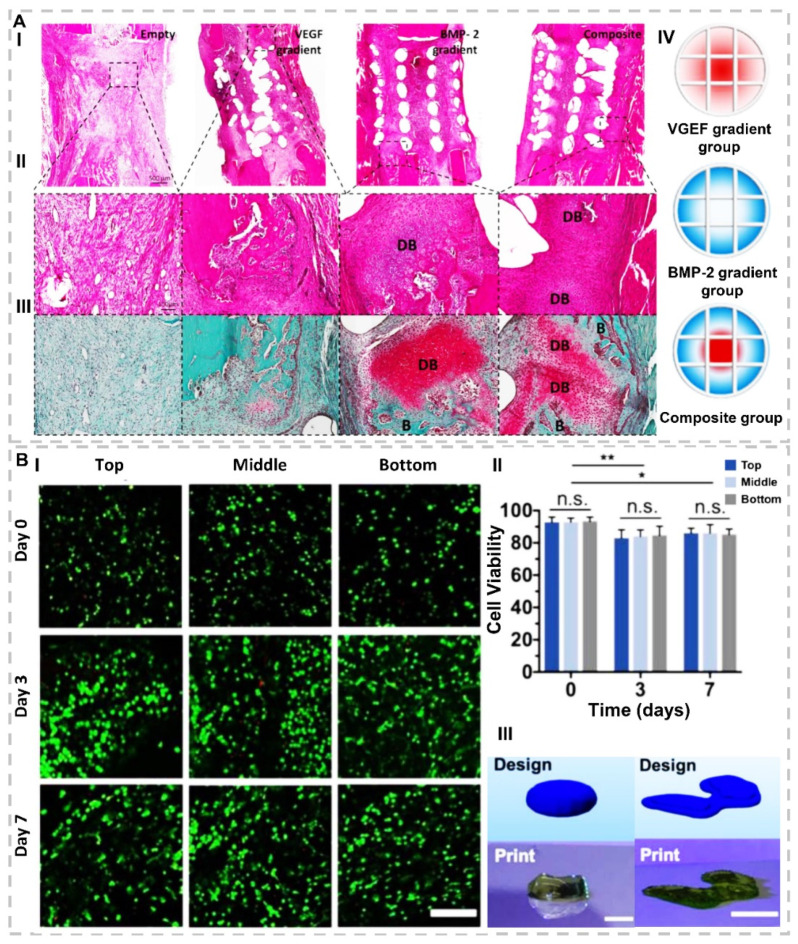
(**A**). Investigation of impact of scaffolds containing BMP-2, VEGF, and their composition on defect regeneration after 2 weeks. (I) H&E stain with 500 µm and (II) with 100 µm scalebar. (III) Safranin O-stained DB points illustrate that cartilage develops to bone by undergoing endochondral ossification, and (**B**) points illustrate new bone tissues. (IV) Schematic of 3D-printed experimental groups, including key features of developed bioinks and segmental defect procedure. Construct design (4 mm in diameter, 5 mm in height)(reproduced content is open access) [48]. (**B**). (I) Representative Live/Dead images with 200 μm scale bar for cell viability and distribution in printed constructs for three layers (top, middle, and bottom) of scaffolds (II) quantification of cell viability, for top, middle, and bottom of printed discs after 0, 3, and 7 days of culture. n ≥ 3, * *p* < 0.05, ** *p* < 0.01, n.s. = not significant. (III) CAD design in comparison to representative image of a printed construct for designs of (right) a model femoral condyle and (left) a disc (~1.5 mm thickness, ~6.5 mm diameter). Scale bars = 1 cm (right) and 5 mm (left)(reproduced content is open access) [65].

Chitosan, derived from chitin, is widely used in bone TE [117,118]. Dong and coworkers formulated PCL-chitosan bioactive ink [66]. The results demonstrated that chitosan as an additive improved osteoinductivity of 3D-printed constructs compared with that of pure PCL ones. Demirtaş et al. [67] examined the effect of chitosan-HAp bioink on morphology, viability, proliferation, and mineralization of preosteoblasts during 21 days compared with that of alginate-HAp inks. Their results illustrated better mechanical properties and cell activities of chitosan-based scaffolds than alginate-based bioinks. In a recent study, the in-vitro expression of chondrogenic markers was evaluated by chondrocytes-loaded ethylenediaminetetraacetic acid (EDTA) modified chitosan bioactive ink [68]. The results illustrated that the chitosan modification did not negatively affect biocompatibility and chondrocytes phenotype while improving printability and shape fidelity.

Mannuronic and guluronic acid form a natural polymer called alginate, which stimulates regeneration in osteochondral defects [119,120,121]. Although alginate supports chondrogenic differentiation, its bioactivity improves through modification [122,123]. One of the main properties of alginate is cross-linking potential with Ba^2+^, Ca^2+^, and Sr^2+^ ions that induce a degree of bioactivity to the prepared structures. A group of researchers used cross-linked alginate as a single component bioink for cartilage TE [69]. The results demonstrated cross-linking process led to higher stability of constructs and the possibility of Ca^2+^ release, which supports chondrogenic differentiation. Nguyen and accomplices indicated the potential of nanofibrillated cellulose (NFC)-alginate bioink for cartilage mimicking when human-derived induced pluripotent stem cells (iPSCs) co-cultured with irradiated chondrocytes [70]. Herein, the bioink formulation provides a nontoxic environment that leads to cell activities and chondrogenic differentiation compared to that of NFC-HA.

Although natural polymers stimulate tissue regeneration due to their molecular domains, they can be combined with other materials such as synthetic polymers and ceramics due to their poor mechanical properties. Some synthetic polymers also accompany surface hydrophobicity and create an unsuitable substrate for cell adhesion, which challenges the formulations of polymeric composite bioactive inks [113].

Bioactive ceramic-containing inks

In the osteochondral TE world, bioceramics are the most widely used biomaterials regarding their bioactivity, bioresorbability, mechanical property, osteoconductivity, and osteogenesis [124]. 

Calcium phosphates are a group of bioceramics containing calcium ions with inorganic phosphate anions. This family can affect osteoblasts’ bioactivity, cell adhesion, proliferation, and new bone formation [125]. In a novel experience, calcium phosphate cement (CPC) and alginate-based bioactive ink were suggested to print the scaffold via an extrusion-based device for osteochondral defects repair, illustrated in Figure 3 [71]. The results indicated an enhancement in mineralization by increasing CPC and proved the potential of pure CPC to simulate the subchondral bone layer. Tricalcium phosphate (TCP) is a bioactive, biocompatible, and biodegradable ceramic supporting bone regeneration. Kim et al. [72] proposed a new bioactive ink formulation based on collagen and TCP to achieve a suitable scaffold for osteogenic differentiation of human adipose stem cells (hASCs). The addition of TCP to ink improved bioactivity, stimulated osteogenesis, and increased printability compared to pure collagen. Koziol and coworkers added TCP nanoparticles to GelMA-alginate inks for regeneration of calcified layer in osteochondral tissue [73]. TCP addition led to appropriate shape fidelity and rheological properties. Additionally, the expression of relevant chondrogenic and osteogenic genes confirmed the supportive behavior of bioactive ink for calcified layer repair.

HAp, the main component in natural bone, is the other calcium phosphates that have found versatile application in bone regeneration [126]. Wenz and colleagues [74] prepared a hASCs-laden bioactive ink consisting GelMA-HAp for bioprinting bone substitutes. The results demonstrated the positive effect of the HAp on osteoconductivity and rheological properties, which facilitate printability and shape fidelity. In a recent study, a new bioink formulation based on gelatin with high weight fraction HAp as a bioactive additive was developed [75]. Here, HAp illustrated hASCs osteogenic differentiation, which arises from its complementary role for collagen in bone TE. A group of scholars evaluated the osteogenic differentiation of hBMSCs in collagen hydrogels containing HAp nanoparticles and deproteinized bovine bone in separated groups [76]. Both 3D-printed scaffolds significantly increased the level of osteogenic-related genes expression compared to that of pure collagen. 

Bioglass (BG) is one of the promising biomaterials for bone regeneration because it stimulates gene expression and osteocalcin production [127]. In an innovative experience, fabricated poly-(hydroxybutyrate-co-hydroxy valerate) (PHBV)-45S5 BG scaffolds by fused deposition modeling (FDM) technique for bone TE [77]. In respect, BG addition improved extrudability and led to superior printability. The interconnected pores with 100–800 µm sizes were obtained by controlling the infill density from 20% to 90%. The tensile modulus of scaffolds was measured from 0.25 to 1.36 GPa, which is similar to the mechanical properties of trabecular bone. The in-vitro results demonstrated no cytotoxicity, and MC3T3-E1 cells (immortalized pre-osteoblastic cells derived from C57BL/6 mouse calvaria) portrayed more spread on BG-containing scaffolds than BG-free scaffolds. Liu et al. [78] developed a biphasic scaffold based on a BG-incorporated hydrogel. The first network ink is composed of glycol chitosan and dibenzaldhyde functionalized polyethylene oxide, and the second is composed of sodium alginate and calcium chloride. Results demonstrated chondrogenic and osteogenic functionalities of constructs as a function of BG addition, which can support osteochondral tissue regeneration in a rabbit model. Distler and colleagues investigated the influence of 45S5 BG on bioactivity, cytocompatibility, and osteoinductivity in polylactic acid (PLA) scaffolds fabricated by the FDM method [79]. Although PLA-BG illustrated a brittle fracture, toughness reduction, and tensile strength reduction by increasing BG content, 1 wt.% BG provided acceptable printability and shape fidelity similar to commercial PLA, which is standard for FDM. In-vitro results evidenced no considerable cytotoxicity but desirable cell viability, making this composition suitable for MC3T3-E1 guidance. Furthermore, gene expression results investigated higher collagen expression and osteocalcin in the presence of BG, which both markers confirmed bioactivity and osteogenic effectiveness of scaffolds (Figure 4). In the other study by researchers, collagen-based inks’ mechanical and stability improvements in BG presence were determined [9]. The results exhibited an improvement in the rheological properties of bioactive ink as a function of BG addition and controlling the structure’s degradation rate. Additionally, stimulating osteogenic differentiation is the most crucial feature arising from BG in the ink composition. Ghorbani and coworkers synthesized PCL/BG ink to fabricate scaffolds with the FDM method [80]. The effects of BG addition were investigated on both mechanical and biological sides. Here, BG addition induced biomineralization of the HAp-like layer after 28 days incubation in SBF solution. The in-vitro results demonstrated more than 90% viability as well as BMSCa differentiation, ALP activation, and both osteopontin and osteocalcin expression for BG-containing scaffolds.

Bioceramics can enrich with bioactive ions, such as Manganese (Mn), Lithium (Li), Silicon (Si), Zinc (Zn), or Strontium (Sr), to increase their effectiveness for tissue regeneration. Printing the Sr_5_(PO_4_)_2_SiO_4_ (SPS) bioactive ink was followed in the investigation by a group of scholars [82]. They observed that the release of Sr and Si ions could significantly stimulate chondrocyte proliferation and suggested their bioactive ink formulation for osteochondral defects repair by its ability to activate the Hypoxia-Inducible Factor (HIF) and Wingless/Int (Wnt) pathways (Figure 5A). Xavier and colleagues fabricated the collagen-nanosilicates bioactive ink to promote osteogenesis due to ultrathin and high anisotropy degree properties of nanosilicate and functionality that lead to surface bio-interactions [83]. Herein, Nanosilicates demonstrated osteogenic promotion without any other osteoinductive stimulator. Increasing pore size, network stiffness, ALP activity, and matrix mineralization are other effects of nanosilicate.

Graphene oxide and carbon nanotube are other nanostructured ceramic types that possibly fabricate bioactive inks. In this regard, Choe et al. [84] added ascending concentration of graphene oxide (0.05, 0.25, 0.5, and 0.1 (mg/mL)) as a bioactive component to alginate bioink for osteogenic regeneration. The proliferation and viability of BMSCs in the presence of graphene oxide were significantly higher than pure alginate. This phenomenon is attributed to antioxidant activity and protein adsorption of graphene oxide by neutralizing toxic reactive oxygen species (ROS) and enhancing cell survival signals in an oxidative stress environment. Bioactive inks containing 0.5 mg/mL graphene oxide illustrated the highest ALP activity, calcium deposition, as well as expression of multiple osteogenic markers in addition to acceptable printability and shape fidelity. A group of researchers investigated the effect of different concentrations of graphene oxide to alginate-gelatin bioactive ink on osteogenic differentiation of hMSCs as well as printability and shape fidelity [85]. The cell-laden scaffolds with 1 mg/mL graphene oxide concentration in bioreactor illustrated better gene expression and ECM mineralization after 42 days than alginate-gelatin scaffolds (Figure 5B). In the case of carbon nanotubes, Huang and coworkers used PCL, HAp, and multiwalled carbon nanotubes (MWCNTs) to fabricate hierarchical 3D-printed scaffolds [86]. Their investigation revealed that additives directly affect mechanical properties and stimulate cellular biological activities such as mineralization, proliferation, and differentiation. The results demonstrated that HAps could improve mineralization, but early-stage differentiation is not the best selection. By adding MWCNTs, osteogenic differentiation increased dramatically; thus, the PCL-HAp-MWCNTs scaffolds exhibited improved ALP, collagen, and osteocalcin expressions compared to PCL or PCL/HAp scaffolds. Cui et al. [87] observed more expression of osteogenesis-related genes by adding MWCNTs to a tough polyion complex (PIC) hydrogel for rat BMSCs. Due to high bone mineral density illustrated after 8 weeks from in-vivo analyses on implanted scaffolds, PIC-MWCNT scaffolds’ potential for promoting bone regeneration and TE applications was confirmed.

In conclusion, bioactive ceramics are among the most widely used biomaterials in biphasic scaffolds and multicomponent bioinks due to their osteoconductivity [128]. Bioactive ceramics successfully stimulate various stem cells’ proliferation, differentiation, and bone TE by reacting and forming chemical bonds with cells and tissues in the biological environment. Many chemical combinations of bioactive ceramics provide a superior foundation for controlling and optimizing biological and physicochemical features. Moreover, bioactive nanoceramics present a significant potential for bony tissues repair than conventional ceramics due to their better biological and mechanical properties [127].

**Figure 5 gels-07-00274-f005:**
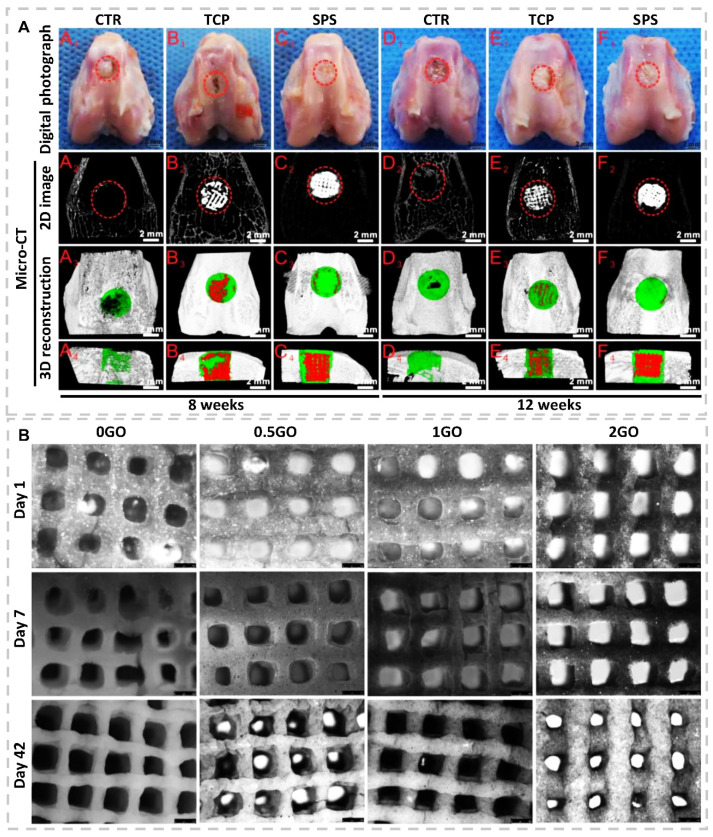
(**A**) Overall photographs and Micro-CT imaging analysis of defects at 8- and 12-weeks postsurgery. (A1–F1) Gross morphology of defects; (A2–F2) 2D projection images of defects; (A3–F3) and (A4–F4) illustrated transverse view and sagittal view of 3D construction images. Off-white color presents primary bone, and white color stands for scaffolds in 2D projection images. Furthermore, off-white color illustrates primary bone, green color illustrates new bone, and red color stands for scaffolds. Compared to that of CTR (blank control) and TCP groups, Micro-CT analysis of defect space exhibited a distinctly greater level of bone regeneration in SPS group (reproduced content is open access) [82]. (**B**) Light microscopy images of 3D bioprinted cell-laden GO scaffolds cultured in osteogenic media for 1, 7, and 42 days (reproduced content is open access) [85].

## 5. Summary and Perspectives

This minireview covered the osteochondral tissue structure, the requirements of bioinks, and the effect of different parameters on printability and shape fidelity. Furthermore, the various categories of bioactive inks, such as growth factor-containing, DNA-containing, ECM-based, bioactive polymer-based, and bioactive ceramics-containing inks, focusing on pieces of research, were described. Overall, in the case of bioactive inks, bioactivity, and biocompatibility, as well as osteogenic and chondrogenic performance, there are clear benefits over traditional bioinks in many studies. Advancements are expected across osteochondral TE, mainly as novel formulation makes diverse inks available to users and as developed research generates further interest for their use.

Increasing the efficacy of osteochondral regeneration requires the multi-material and stratified constructs implanted in the defective site. In detail, novel bioprinting technology can reach hierarchical constructs with different layers’ properties. Additionally, bioactive materials in the chemical composition of scaffolds can guide tissue-scaffold interactions and bind with surrounding tissues, as well as control the body’s immune response, which follows the success of an implant. However, printability and shape fidelity of formulated bioactive inks are crucial parameters as well as biological properties, which should be considered by controlling viscosity, shear-thinning, viscoelasticity, and yield stress. 

Recently, there were pieces of research on the development of bioprinting scaffolds by bioactive inks containing bioactive components affecting osteochondral regeneration. The investigations demonstrated that formulating bioactive inks is a promising strategy for bone and cartilage regeneration. However, the limiting factors make progress in this field challenging. Although bioactive components such as bioactive ions or growth factors in cell-laden bioactive inks improve the biofunctionality and regeneration for osteochondral tissue, they are commercially useless due to limitations such as low cell survival rate and growth factors instability. Besides, not only the printing parameters should not damage the bioactive ink and cells, but also the bioink should not disrupt the printing process and maintains the structure of the final scaffold. On the other hand, both the bioink and the bioprinting method must provide the desired properties and shape for the complex osteochondral tissue, considering the different physiological properties of the osteochondral tissue layers and stimulating tissue regeneration. If bioactive inks formulations promote preferred osteochondral regeneration to exist strategies and afford the means to limit and treat diseases such as osteoarthritis, they will vindicate attempting to overwhelm the many managerial and commercial challenges that will be confronted with the clinical development of bioactive inks outcomes. On this point, the research path in this field continues to achieve an ideal formulation that does not activate the body’s immune and inflammatory responses and increases the speed and quality of osteochondral tissue regeneration in clinical approaches.

## Figures and Tables

**Figure 1 gels-07-00274-f001:**
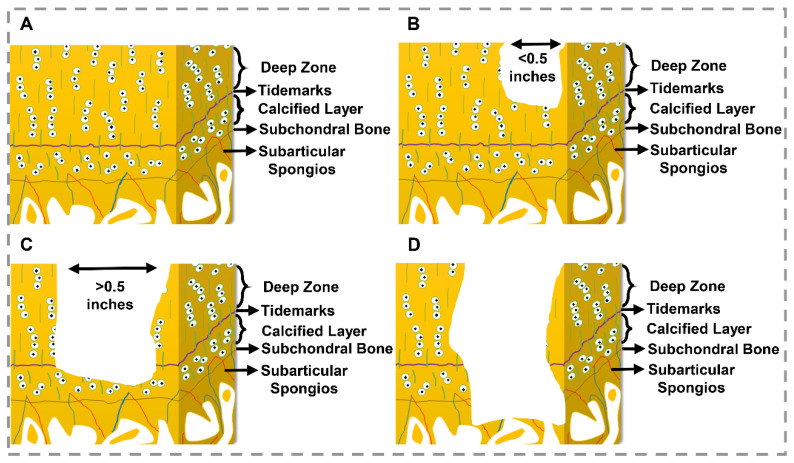
Osteochondral defect grades. (**A**) Normal cartilage (grade 0), (**B**) Moderate lesion (grade I: partial-thickness defects), (**C**) Sever lesion (grade III: full-thickness defects), (**D**) Very sever lesion (grade IV: osteochondral defect) [24,30,31].

**Figure 3 gels-07-00274-f003:**
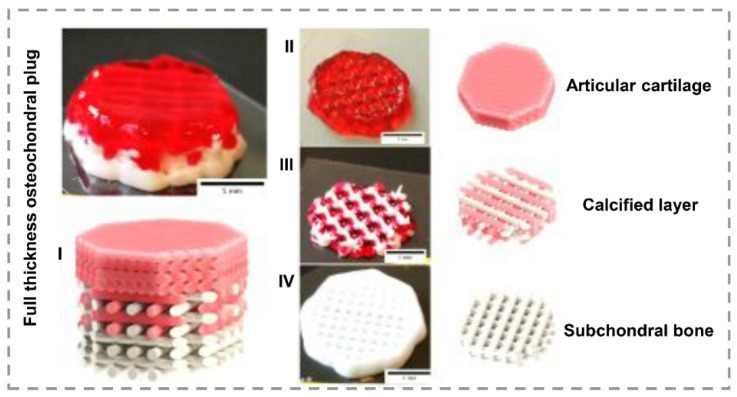
(I) multilayered, 3D-bioprinted scaffold for osteochondral defect. (II) first layer based on alginate ink for articular cartilage regeneration. (III) second layer combination of Alg and CPC for calcified stimulation. (IV) third layer pure CPC for subchondral bone repair. Scale bar illustrates 5mm (reproduced content is open access) [71].

**Figure 4 gels-07-00274-f004:**
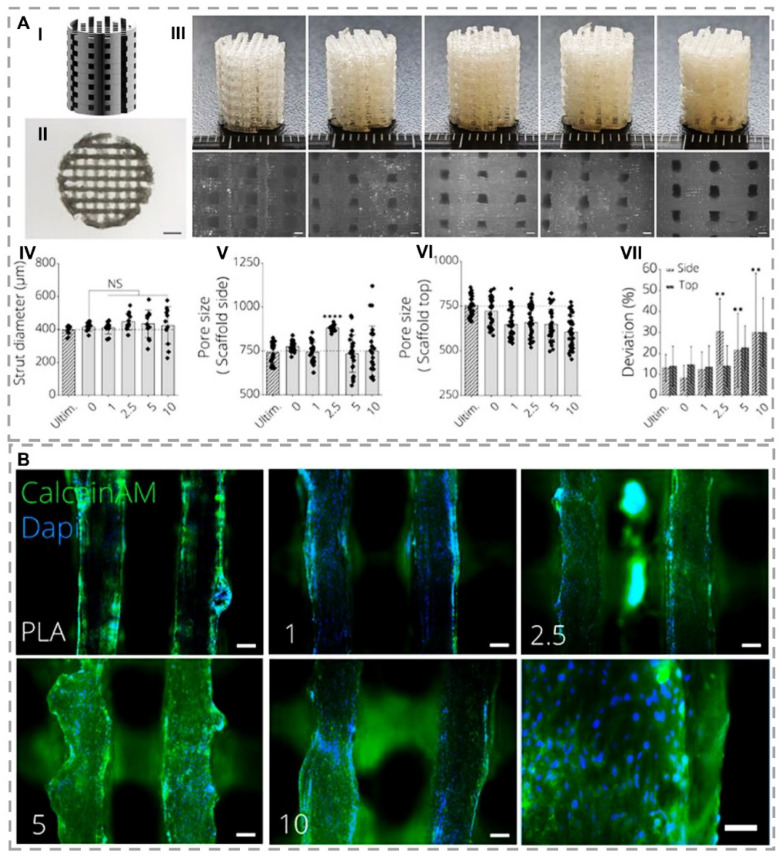
(**A**) Mechanical properties of 3D-printed PLA-45S5 BG scaffolds. (I) designed scaffold using CAD with pore sizes in 750 μm width. (II) Top view image of a PLA-BG scaffold using light microscopy with scale bar = 2 mm. (III) 3D-printed scaffold with FDM method with 0, 1, 2.5, 5, and 10 wt.% BG component (respectively from left to right). Bottom row displays light microscopy images with scale bars = 500 μm. (IV-VII) PLA-BG scaffolds printability and porosity assessment in comparison by Ultimaker PLA filaments as reference depicting (IV) strut diameter, (V) scaffold porosity at side, and (VI) top, as well as (VII) deviation of pore area from theoretical pore area calculated from CAD model as a measure of printing accuracy. (** *p* < 0.01, **** *p* < 0.0001 indicate a statistically significant difference of means compared to 3D-printed 0% BG-PLA by one-way ANOVA or Welch’s *t*-test in pairwise comparisons of scaffold side pore diameter). (**B**) Immunohistochemistry of MC3T3E1 preosteoblast cells cultured on 3D-printed PLA-BG scaffolds after 24 h. Fluorescence microscopy images display Calcein AM (green) and DAPI (blue) stained cells on PLA-BG scaffolds with BG concentration of (I) 0, (II) 1, (III) 2.5, (IV) 5, and (V) 10 wt.% with 100 μm scale bars sizes and (VI) 10 wt.% with 200 μm scale bar size (reproduced content is open access) [79].

**Table 1 gels-07-00274-t001:** Summary of bioactive inks for bone, osteochondral, and cartilage TE.

Matrix Composition	Properties of Adding Bioactive Component	Tissue Target	3D Printing Method	Ref.
Growth Factor-Containing Inks				
Alginate-GelMA- TGF-β3	Promotes ECM Deposition	Cartilage	Extrusion	[47]
Alginate-BMP-2-VEGF	Improving Angiogenesis or Osteogenesis	Bone	Extrusion	[48]
DermaMatrix-BMP-2-noggin	Improving Osteogenic Differentiation	Bone	Inkjet	[49]
Alginate-Gelatin Microparticles-BMP-2	Improving Osteogenesis and Promoting Bone Regeneration	Bone	Extrusion	[50]
PCL-Alginate-BMP-2	Higher GAGs, DNA, and Collagen Content	Cartilage	Extrusion	[51]
Aptamer-TGF-β3-Decellularize ECM-GelMa-PCL	More Chondrogenic Promoting	Cartilage	Extrusion	[52]
DNA-Containing Inks				
Alginate-Methylcellulose -pDNA	Osteogenic and Chondrogenic Differentiation- Bone and Cartilage Formation	Osteochondral	Extrusion	[53]
Polypeptide-DNA	Cell Viability-Structural Stability	-	Extrusion	[54]
Alginate-pDNA	Providing Tissue Access to BMP-2 Genes Which Leads to Osteogenic Differentiation	Bone	Extrusion	[55]
Alginate-Nano HAp-pDNA	Providing Tissue Access to BMP-2 and TGF-β3 Genes Which Leads to Osteogenic Differentiation	Bone	Extrusion	[56]
ECM-Based Inks				
Silk-decellularized ECM	Chondrogenic Differentiation	Cartilage	Stereolithography	[57]
Cartilage decellularized ECM	Better Load Bearing-Chondrogenic Differentiation- Better Printability	Cartilage	Laser	[58]
PEGDA-decellularized ECM	Chondrogenic Promotion with Subchondral Bone Regeneration	Osteochondral	Stereolithography	[59]
Alginate-Collagen-ECM	Providing Cell Activities and Promoting Osteogenic Differentiation	Bone	Extrusion	[60]
Bioactive Polymer-Based Inks				
Alginate-Collagen	Chondrocyte Phenotype Maintenance and Chondrogenic Promotion	Cartilage	Extrusion	[61]
Agarose-Collagen	Osteogenic Differentiation	Bone	Inkjet	[62]
GelMA-HAp	Processability-Good Mechanical properties-Similarity with ECM	Osteochondral	Extrusion	[63]
Alginate-HA	Promoting Chondrogenesis	Cartilage	Extrusion	[64]
HA	Chondrogenic Differentiation	Cartilage	FDM	[65]
PCL-Chitosan	Improving Osteoinductivity	Bone	Extrusion	[66]
Chitosan-HAp	Influence on Morphology, Viability, Proliferation, and Mineralization	Bone	Extrusion	[67]
Chitosan-EDTA	Osteogenic differentiation supporting	Bone	Extrusion	[68]
Alginate	Chondrogenic differentiation by Ca^2+^ release	Cartilage	Extrusion	[69]
NFC-Alginate	Stimulating Proteoglycans-Supporting Chondrogenic Differentiation	Cartilage	Inkjet	[70]
Bioactive Ceramic-Containing Inks				
Alginate-CPC	Increasing Mineralization and Supporting Subchondral Bone Regeneration	Osteochondral	Extrusion	[71]
Collagen-TCP	Improving bioactivity-Stimulating Osteogenesis and Increasing Printability	Bone	Extrusion	[72]
GelMA-Alginate-TCP	Improving Osteogenic and Chondrogenic Differentiation in Addition to Calcified Layer Formation	Osteochondral	Extrusion	[73]
GelMA-HAp	Positive Effect on Osteoconductivity and Rheological Properties	Bone	Extrusion	[74]
Gelatin-HAp	Supporting Osteogenic Differentiation	Bone	Extrusion	[75]
Collagen-HAp	Increase in Osteogenesis-Related Genes Expression	Bone	Extrusion	[76]
PHBV-45S5 BG	Improving Rheological Properties and Cells Spreading	Bone	FDM	[77]
Alginate-Chitosan-BG	Osteoenic and Chondrogenic promotion	Osteochondral	Extrusion	[78]
PLA-BG	Bioactivity, Cytocompatibility, and Osteoinductivity	Bone	FDM	[79]
Collagen-BG	Osteogenic Differentiation in Addition to Improving Rheological Properties	Bone	Extrusion	[9]
PCL-BG	HAp-Like Layer Mineralization, ALP Activation, Osteopontin, and Osteocalcin Expression	Bone	FDM	[80]
Alginate-Gelatin-BG	Higher Mechanical Properties- Higher Cell Viability	Bone	Extrusion	[81]
Alginate-Sr_5_(PO_4_)_2_SiO_4_	Stimulate Chondrocyte Proliferation, Activating the HIF and Wnt Pathways.	Osteochondral	Extrusion	[82]
GelMA-Nanosilicate	Increasing stiffness-Increasing Enzymatic Stability- Improving Tunable Mechanical Properties-Improving Degradation rate-Supporting Osteogenic Differentiation	Bone	Extrusion	[83]
Alginate-Graphene Oxide	Antioxidant Activity-Protein Adsorption-ALP Activity-Calcium Deposition-Osteogenic Markers Expression-Printability-Shape fidelity	Bone	Extrusion	[84]
Alginate-Gelatin-Graphene Oxide	Osteogenic Differentiation and ECM Mineralization	Bone	Extrusion	[85]
PCL-HAp-MWCNTs	Increasing Mineralization, Proliferation, and Differentiation	Bone	Extrusion	[86]
PIC-MWCNT	Osteogenic Differentiation and High Bone Mineral Density	Bone	Extrusion	[87]

## Data Availability

Not applicable.

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
