# Peer review of "Bioactive Inks Development for Osteochondral Tissue Engineering: A Mini-Review"

_gels, 2021, doi:10.3390/gels7040274_

Round 1

Reviewer 1 Report

In the present paper the authors focused on osteochondral tissue and osteochondral scaffolds, which play an influential role in osteochondral regeneration and can provide potential treatment options in 
early-stage osteoarthritis to delay or avoid the use of joint replacements.

Overall the review is well written and the topic is interesting, but still is necessary a major revision. 

major concerns:

The figure 2 need to be revised, the images are out of focus, he col I IH looks overexposed need to be revised. moreover, the figure 5 H&E need to be revised.

the english need to be revised by a native english speaker. 

minor concerns

uM please revised in  µm

Author Response

Reviewer 1:

In the present paper the authors focused on osteochondral tissue and osteochondral scaffolds, which play an influential role in osteochondral regeneration and can provide potential treatment options in early-stage osteoarthritis to delay or avoid the use of joint replacements.Overall the review is well written and the topic is interesting, but still is necessary a major revision. 

  1. The figure 2 need to be revised, the images are out of focus, he col I IH looks overexposed need to be revised. moreover, the figure 5 H&E need to be revised.

Answer: Dear reviewer, thanks for your careful and helpful comment. The figures have been revised according to your valuable suggestion. As the manuscript focused on the bioactivity of the ink, we tried showing the experimental result in addition to the engineering phase. All the changes have been highlighted in the main text.

  1. the english need to be revised by a native english speaker. 

Answer: Dear reviewer, thanks for your attention. Language is edited, and all the changes are highlighted in the main text.

  1. uM please revised in µm

Answer: Dear reviewer, thanks for your kind attention. It is modified in the main text and highlighted.

Reviewer 2 Report

The manuscript reviews the advances in ink materials for osteochondral tissue engineering. The manuscript briefly describes osteochondral defects and the need for tissue engineering applications. The rest of the manuscript is focused on the reviewing the literature on the biomaterials used for developing inks for osteochondral tissue engineering.

This is an important medical challenge and there are several research groups studying the problem. Therefore, a proper review paper could be exciting for researchers and clinicians in the field. However, the manuscript stays as a literature survey, rather than a critical review. For example, the benefits of the materials and their limitations, or different biological factors are not properly discussed. It is not clear if any of the technologies has the potential for treating osteochondral defects and what are the limitations that should be overcome. So, I do not recommend its publication in the current form. The following are some specific comments:

  • Page 6, some of the listed growth factors are neither osteogenic nor condrogenic.
  • The section dedicated to nucleic acid based delivery is very superficial.
  • There is not critical conclusion on the pros and cons of each material category.
  • What is the size and dimensions of osteochondoral defects and how those can be reconstructed using bioprinting. Are the bioinks sufficient for achieving this goal.
  • The section on limitations and future outlooks is not sufficiently developed.
  • What are the suitable strategies for the delivery of inks for treatment of osteochondral defects. For example, portable or handheld bioprinters for in situ printing of scaffolds can be discussed as a potential strategy for proper implantation of the scaffolds.

Author Response

Reviewer 2:

The manuscript reviews the advances in ink materials for osteochondral tissue engineering. The manuscript briefly describes osteochondral defects and the need for tissue engineering applications. The rest of the manuscript is focused on the reviewing the literature on the biomaterials used for developing inks for osteochondral tissue engineering. This is an important medical challenge and there are several research groups studying the problem. Therefore, a proper review paper could be exciting for researchers and clinicians in the field. However, the manuscript stays as a literature survey, rather than a critical review. For example, the benefits of the materials and their limitations, or different biological factors are not properly discussed. It is not clear if any of the technologies has the potential for treating osteochondral defects and what are the limitations that should be overcome. So, I do not recommend its publication in the current form. The following are some specific comments:

  1. Page 6, some of the listed growth factors are neither osteogenic nor chondrogenic.

Answer: Dear reviewer, we greatly appreciate your thoughtful comments to improve the manuscript. Briefly, a variety of growth factors support osteogenic and chondrogenic differentiation as well as regulating osteoblasts’ activity for bone regeneration. For example, bone morphogenetic protein (BMP) (2, 4, 6, and 7) or members of transforming growth factor-β (TGF-β) family, support chondrocyte and osteogenic differentiation, bone and fracture healing, bone and cartilage development, bone marrow formation, etc. [1–5]. Alternatively, fibroblast growth factor-2 (FGF-2) promotes bone development and expresses in the endochondral bone formation stage as well as regulating chondrocytes differentiation and proliferation [6]. Insulin-like growth factor (IGF) (1, 2) is another influential growth factor in osteogenic and chondrogenic differentiation [4,7–12]. Additionally, platelet-derived growth factor (PDGF) and vascular endothelial growth factor (VEGF) have demonstrated synergistic effects besides the factors mentioned above to increase osteogenic and chondrogenic differentiation and tissue regeneration [13–18].

We carefully revised the manuscript according to your comments, and all the changes have been highlighted in the main text.

  1. The section dedicated to nucleic acid based delivery is very superficial

Answer: Thank you for considering this issue. "DNA-containing inks" section was modified, and all the changes have been highlighted in the main text. Briefly,  DNA hydrogels are biodegradable, designable, and permeable compositions to construct ingenious TE patterns [19]. Therapeutic proteins expression can stimulate tissue formation following ECM production, and this approach is achievable by non-viral gene delivery. In this regard, a gene activated inks provide a promising approach in biofabrication strategies and biological functionality of scaffolds [20]

  1. There is not critical conclusion on the pros and cons of each material category.

Answer: Many thank you for your attention. It has been edited according to your suggestion. All the changes have been highlighted in the main text.

  1. What is the size and dimensions of osteochondoral defects and how those can be reconstructed using bioprinting. Are the bioinks sufficient for achieving this goal.

Answer: We greatly appreciate your thoughtful comments that helped to improve the manuscript. Based on the outerbridge classification system, there are 5 classification for osteochondral tissue: 0) normal cartilage, 1) mild lesion with just softening and swelling in cartilage, 2) moderate lesion with partial-thickness and width less than 0.5 inches, 3) severe lesion with full-thickness and width more than 0.5 inches, and 4) very severe lesion with subchondral bone damage, which means involvement of the entire thickness of the osteochondral tissue, about 2 to 4 mm [21–24]. Such damages can be repaired using multi-component hierarchical and stratified scaffolds [25]. Due to the possibility of mimicking the gradient structure and properties of the osteochondral unit, bioprinting is a suitable method for personalized osteochondral tissue regeneration and translational medicine. The bioprinting method makes it possible to fabricate multi-layer and multi-material precise structures, resembling defect size and shape, by using 3D imaging such as MRI (magnetic resonance imaging) or CT scan (computed tomography) and processing in CAD (computer-aided design) software [26]. In this regard, bioactive inks are an excellent choice for printing the reconstructive constructs because of accelerating regeneration through different formulations using growth factors, cells, and bioactive substances.

We revised the manuscript according to your comments, and all the changes have been highlighted in the main text.

  1. The section on limitations and future outlooks is not sufficiently developed.

Answer: Thank you so much. This section have been edited. All the changes have been highlighted in the main text. Briefly, the limiting factors make progress in this field challenging. Although bioactive components such as bioactive ions or growth factors in cell-laden bioactive inks improve the biofunctionality and regeneration for osteochondral tissue, they are commercially useless due to limitations such as low cell survival rate and growth factors instability. Besides, the bioink should not disrupt the printing process and maintains the structure of the final scaffold; in addition, printing parameters should not damage the bioactive ink and cells. On the other hand, both the bioink and the bioprinting method must provide the desired properties and shape for the complex osteochondral tissue, considering the different physiological properties of the osteochondral tissue layers, and stimulate tissue regeneration. If bioactive inks formulations promote preferred osteochondral regeneration to exist strategies and afford the means to limit and treat diseases such as osteoarthritis, they will vindicate attempting to overwhelm the many managerial and commercial challenges that will be confronted with the clinical development of bioactive inks outcomes.

  1. What are the suitable strategies for the delivery of inks for treatment of osteochondral defects. For example, portable or handheld bioprinters for in situ printing of scaffolds can be discussed as a potential strategy for proper implantation of the scaffolds.

Answer: Dear reviewer, thank you for taking the time to review our work. Briefly, there are different classifications of bioprinting methods for osteochondral tissue regeneration. However, technically the three main bioprinting methods, including (1) inkjet-based, (2) laser-based, and (3) extrusion-based bioprinting, have gained attention, considering bioink formula and required microstructure [27]. Herein, bioprinted scaffolds with the specific defect shape are implanted at the defect site and stimulate regeneration in the body's environment. In a recent study a bioactive bilayer bioprinted scaffold based on silk fibroin and decellularized extracellular matrix riched with osteogenic growth factors implanted in the rabbits’ knee joint model for osteochondral regeneration [28]. Besides, in-situ bioprinters that include a robotic arm and a handheld approach have shown particular application in bone and cartilage TE [29]. Hereof, in an innovative experience, in-vivo hyaline-like cartilage formation was observed using in-situ bioprinting with coaxial nozzle extrusion-based technique based on gelatin methacrylate (GelMA) and hyaluronic acid (HA) [30]. Additionally, Keriquel and co-workers explored in-situ bone formation after implantation of a laser-based bioprinted scaffold based on collagen and hydroxyapatite (HAp) in a calvaria defect in a mice model [27]. All the changes have been highlighted in the main text.

References

  1. Mizrahi, O.; Sheyn, D.; Tawackoli, W.; Kallai, I.; Oh, A.; Su, S.; Da, X.; Zarrini, P.; Cook-Wiens, G.; Gazit, D.; et al. BMP-6 is more efficient in bone formation than BMP-2 when overexpressed in mesenchymal stem cells. Gene Ther. 2013, 20, 370–377, doi:10.1038/gt.2012.45.
  2. Itoh, F. Promoting bone morphogenetic protein signaling through negative regulation of inhibitory Smads. EMBO J. 2001, 20, 4132–4142, doi:10.1093/emboj/20.15.4132.
  3. Garrison, K.R.; Shemilt, I.; Donell, S.; Ryder, J.J.; Mugford, M.; Harvey, I.; Song, F.; Alt, V. Bone morphogenetic protein (BMP) for fracture healing in adults. Cochrane Database Syst. Rev. 2010, doi:10.1002/14651858.CD006950.pub2.
  4. Mariani, E.; Pulsatelli, L.; Facchini, A. Signaling Pathways in Cartilage Repair. Int. J. Mol. Sci. 2014, 15, 8667–8698, doi:10.3390/ijms15058667.
  5. Derynck, R.; Zhang, Y.E. Smad-dependent and Smad-independent pathways in TGF-β family signalling. Nature 2003, 425, 577–584, doi:10.1038/nature02006.
  6. Kronenberg, H.M. Developmental regulation of the growth plate. Nature 2003, 423, 332–336, doi:10.1038/nature01657.
  7. Guntur, A.R.; Rosen, C.J. IGF-1 regulation of key signaling pathways in bone. Bonekey Rep. 2013, 2, doi:10.1038/bonekey.2013.171.
  8. Pasold, J.; Bader, R.; Zander, K.; Heskamp, B.; Grüttner, C.; Lüthen, F.; Tischer, T.; Jonitz-Heincke, A. Positive impact of IGF-1-coupled nanoparticles on the differentiation potential of human chondrocytes cultured on collagen scaffolds. Int. J. Nanomedicine 2015, 1131, doi:10.2147/IJN.S72872.
  9. Mullen, L.M.; Best, S.M.; Ghose, S.; Wardale, J.; Rushton, N.; Cameron, R.E. Bioactive IGF-1 release from collagen–GAG scaffold to enhance cartilage repair in vitro. J. Mater. Sci. Mater. Med. 2015, 26, 2, doi:10.1007/s10856-014-5325-y.
  10. Starkman, B.G.; Cravero, J.D.; Delcarlo, M.; Loeser, R.F. IGF-I stimulation of proteoglycan synthesis by chondrocytes requires activation of the PI 3-kinase pathway but not ERK MAPK. Biochem. J. 2005, 389, 723–729, doi:10.1042/BJ20041636.
  11. Salgado, A.J.; Coutinho, O.P.; Reis, R.L. Bone Tissue Engineering: State of the Art and Future Trends. Macromol. Biosci. 2004, 4, 743–765, doi:10.1002/mabi.200400026.
  12. Deng, C.; Chang, J.; Wu, C. Bioactive scaffolds for osteochondral regeneration. J. Orthop. Transl. 2019, 17, 15–25, doi:10.1016/j.jot.2018.11.006.
  13. Castro, P.R.; Marques, S.M.; Viana, C.T.R.; Campos, P.P.; Ferreira, M.A.N.D.; Barcelos, L.S.; Andrade, S.P. Deletion of the chemokine receptor CCR2 attenuates foreign body reaction to implants in mice. Microvasc. Res. 2014, 95, 37–45, doi:10.1016/j.mvr.2014.07.002.
  14. Wang, J.; Liu, D.; Guo, B.; Yang, X.; Chen, X.; Zhu, X.; Fan, Y.; Zhang, X. Role of biphasic calcium phosphate ceramic-mediated secretion of signaling molecules by macrophages in migration and osteoblastic differentiation of MSCs. Acta Biomater. 2017, 51, 447–460, doi:10.1016/j.actbio.2017.01.059.
  15. Chung, E.J.; Chien, K.B.; Aguado, B.A.; Shah, R.N. Osteogenic Potential of BMP-2-Releasing Self-Assembled Membranes. Tissue Eng. Part A 2013, 19, 2664–2673, doi:10.1089/ten.tea.2012.0667.
  16. Stavroulaki, E.; Kastrinaki, M.-C.; Pontikoglou, C.; Eliopoulos, D.; Damianaki, A.; Mavroudi, I.; Pyrovolaki, K.; Katonis, P.; Papadaki, H.A. Mesenchymal Stem Cells Contribute to the Abnormal Bone Marrow Microenvironment in Patients with Chronic Idiopathic Neutropenia by Overproduction of Transforming Growth Factor-β1. Stem Cells Dev. 2011, 20, 1309–1318, doi:10.1089/scd.2010.0425.
  17. Toosi, S.; Behravan, J. Osteogenesis and bone remodeling: A focus on growth factors and bioactive peptides. BioFactors 2020, 46, 326–340, doi:10.1002/biof.1598.
  18. Zhou, M.; Lozano, N.; Wychowaniec, J.K.; Hodgkinson, T.; Richardson, S.M.; Kostarelos, K.; Hoyland, J.A. Graphene oxide: A growth factor delivery carrier to enhance chondrogenic differentiation of human mesenchymal stem cells in 3D hydrogels. Acta Biomater. 2019, 96, 271–280, doi:10.1016/j.actbio.2019.07.027.
  19. Li, C.; Faulkner-Jones, A.; Dun, A.R.; Jin, J.; Chen, P.; Xing, Y.; Yang, Z.; Li, Z.; Shu, W.; Liu, D.; et al. Rapid Formation of a Supramolecular Polypeptide-DNA Hydrogel for In Situ Three-Dimensional Multilayer Bioprinting. Angew. Chemie 2015, 127, 4029–4033, doi:10.1002/ange.201411383.
  20. Cunniffe, G.M.; Gonzalez-Fernandez, T.; Daly, A.; Sathy, B.N.; Jeon, O.; Alsberg, E.; Kelly, D.J. Three-Dimensional Bioprinting of Polycaprolactone Reinforced Gene Activated Bioinks for Bone Tissue Engineering. Tissue Eng. Part A 2017, 23, 891–900, doi:10.1089/ten.tea.2016.0498.
  21. Sophia Fox, A.J.; Bedi, A.; Rodeo, S.A. The Basic Science of Articular Cartilage: Structure, Composition, and Function. Sport. Heal. A Multidiscip. Approach 2009, 1, 461–468, doi:10.1177/1941738109350438.
  22. Mueller-gerbl, M. The basic science of the subchondral bone. 2010, 419–433, doi:10.1007/s00167-010-1054-z.
  23. Herrero-beaumont, G.M.D. Correlation between arthroscopic and histopathological grading systems of articular cartilage lesions in knee osteoarthritis. Osteoarthr. Cartil. 2009, 17, 205–212, doi:10.1016/j.joca.2008.06.010.
  24. Slattery, C.; Kweon, C.Y. Classifications in Brief: Outerbridge Classification of Chondral Lesions. Clin. Orthop. Relat. Res. 2018, 476, 2101–2104, doi:10.1007/s11999.0000000000000255.
  25. Tamaddon, M.; Wang, L.; Liu, Z.; Liu, C. Osteochondral tissue repair in osteoarthritic joints: clinical challenges and opportunities in tissue engineering. Bio-Design Manuf. 2018, 1, 101–114, doi:10.1007/s42242-018-0015-0.
  26. Seol, Y.-J.; Kang, H.-W.; Lee, S.J.; Atala, A.; Yoo, J.J. Bioprinting technology and its applications. Eur. J. Cardio-Thoracic Surg. 2014, 46, 342–348, doi:10.1093/ejcts/ezu148.
  27. Keriquel, V.; Oliveira, H.; Rémy, M.; Ziane, S.; Delmond, S.; Rousseau, B.; Rey, S.; Catros, S.; Amédée, J.; Guillemot, F.; et al. In situ printing of mesenchymal stromal cells, by laser-assisted bioprinting, for in vivo bone regeneration applications. Sci. Rep. 2017, 7, 1778, doi:10.1038/s41598-017-01914-x.
  28. Zhang, X.; Liu, Y.; Zuo, Q.; Wang, Q.; Li, Z.; Yan, K.; Yuan, T.; Zhang, Y.; Shen, K.; Xie, R.; et al. 3D Bioprinting of Biomimetic Bilayered Scaffold Consisting of Decellularized Extracellular Matrix and Silk Fibroin for Osteochondral Repair. Int. J. Bioprinting 2021, 7, 401, doi:10.18063/ijb.v7i4.401.
  29. Singh, S.; Choudhury, D.; Yu, F.; Mironov, V.; Naing, M.W. In situ bioprinting – Bioprinting from benchside to bedside? Acta Biomater. 2020, 101, 14–25, doi:10.1016/j.actbio.2019.08.045.
  30. Di Bella, C.; Duchi, S.; O’Connell, C.D.; Blanchard, R.; Augustine, C.; Yue, Z.; Thompson, F.; Richards, C.; Beirne, S.; Onofrillo, C.; et al. In situ handheld three‐dimensional bioprinting for cartilage regeneration. J. Tissue Eng. Regen. Med. 2018, 12, 611–621, doi:10.1002/term.2476.

Round 2

Reviewer 2 Report

The revised manuscript has improved significantly and the authors have addressed my concerns. I recommend the publication of the manuscript in the current form.